# Ultrahigh high-strain-rate superplasticity in a nanostructured high-entropy alloy

Nhung Thi-Cam Nguyen[1,2,4], Peyman Asghari-Rad [1,2,4], Praveen Sathiyamoorthi [1,2], Alireza Zargaran[3], Chong Soo Lee [3] & Hyoung Seop Kim [1,2,3✉]

Superplasticity describes a material's ability to sustain large plastic deformation in the form of a tensile elongation to over 400% of its original length, but is generally observed only at a low strain rate ($\sim 10^{-4}\,s^{-1}$), which results in long processing times that are economically undesirable for mass production. Superplasticity at high strain rates in excess of $10^{-2}\,s^{-1}$, required for viable industry-scale application, has usually only been achieved in low-strength aluminium and magnesium alloys. Here, we present a superplastic elongation to 2000% of the original length at a high strain rate of $5 \times 10^{-2}\,s^{-1}$ in an $Al_9(CoCrFeMnNi)_{91}$ (at%) high-entropy alloy nanostructured using high-pressure torsion. The high-pressure torsion induced grain refinement in the multi-phase alloy combined with limited grain growth during hot plastic deformation enables high strain rate superplasticity through grain boundary sliding accommodated by dislocation activity.

[1] Department of Materials Science and Engineering, Pohang University of Science and Technology (POSTECH), Pohang 37673, South Korea. [2] Center for High Entropy Alloys, Pohang University of Science and Technology (POSTECH), Pohang 37673, South Korea. [3] Graduate Institute of Ferrous Technology, Pohang University of Science and Technology (POSTECH), Pohang 37673, South Korea. [4] These authors contributed equally: Nhung Thi-Cam Nguyen, Peyman Asghari-Rad. ✉email: hskim@postech.ac.kr

Superplasticity is generally observed in fine-grained materials at low strain rates ($10^{-4}$–$10^{-3}$ s$^{-1}$) and high homologous temperatures (>0.5 $T_m$), where $T_m$ is the melting temperature[1-3]. Over the past few decades, achieving superplasticity at high-strain rates (>$10^{-2}$ s$^{-1}$) has garnered wide attention from researchers as it possesses great advantage of reducing the forming time, making it economically viable for industries[4,5]. However, High Strain-Rate Superplasticity (HSRS) is extremely rare in high-strength materials, and in particular, the lack of superplastic forming capability in a recently emerged class of high-strength materials, the so-called High-Entropy Alloys (HEAs), is a serious obstacle for its potential use in engineering complex structures. HEAs have a unique alloy design concept based on multi-principle elements, and they exhibit remarkable properties such as high strength combined with high ductility as well as high fracture toughness when compared to conventional alloys[6-8]. Thus, achieving HSRS in HEAs would signify a huge breakthrough in advanced material science.

According to the constitutive equation for superplasticity, a decrease in grain size is essential to achieve superplasticity at a high-strain rate[4,5,9,10]. Besides, Grain Boundary Sliding (GBS), the predominant deformation mechanism found in superplastic behavior, is predominantly observed in materials with small grains and high-angle grain boundaries[11,12]. More importantly, retention of Ultra-Fine Grains (UFGs) or nanoscale grains without significant grain growth at elevated temperatures is essential to accommodate GBS and GBS-induced cavitation through diffusion or dislocation motion[11,13,14]. The representative CoCrFeMnNi HEA with a single Face-Centered Cubic (FCC) structure exhibited a reasonable superplastic elongation of 570% at elevated temperature[15]. There is a consensus that a multi-phase structure potentially shows higher elongation as compared with a single-phase structure due to grain growth suppression during the superplasticity test[1,10]. In most of the HEAs, the addition of Al has been demonstrated to form an ordered Body-Centered Cubic (B2) phase and it is shown to enhance the room temperature mechanical properties[16-18]. It has been reported that the B2 phase in Al$_x$(CoCrFeMnNi)$_{100-x}$ forms when the Al content is above 8 at%, and a drastic decrease in ductility is observed when the Al content is more than 11 at%[19]. Consequently, in the present study, we chose Al$_9$(CoCrFeMnNi)$_{91}$ HEA based on the abovementioned logical reasoning and engineered a unique nanostructured dual-phase microstructure.

Here, we fabricate an Al$_9$(CoCrFeMnNi)$_{91}$ HEA with nanosized FCC grains and B2 phase in the range of a few hundred nanometers to micrometers through thermo-mechanical treatment followed by high-pressure torsion (HPT) processing. Among several processing routes, the HPT is the most effective route to produce bulk materials with very fine grain size[20-22]. The presence of B2 phase and the newly formed sigma phase during superplastic testing assist in preventing significant grain growth, favoring GBS with high-strain-rate sensitivity value. The GBS is observed to be accommodated by interlinkage of cavities, and dislocation activities in FCC and B2 phases. The fine grain size accompanied by high-strain-rate sensitivity and GBS accommodation by plastic deformation lead to an ultra-high superplasticity of 2000% at a high-strain rate of $5 \times 10^{-2}$ s$^{-1}$ in the present alloy. Considering the unique properties reported in HEAs and the HSRS achieved in the present work, HEAs show the potential to be an emerging advanced materials for future structural applications with complex shapes.

## Results and discussion

**Initial microstructure**. The initial microstructure shows FCC and Al–Ni-rich B2 phases before superplastic tensile tests

(Fig. 1). The as-annealed specimen displays (Fig. 1a) fine FCC grains (average size of ~2 μm) with relatively large B2 precipitates (800 nm–1 μm) at the boundaries of FCC grains, and small B2 precipitates inside the FCC grains (200–400 nm)[17,23,24]. The X-Ray Diffraction (XRD) patterns of the as-annealed and as-HPT specimens show the peaks of FCC and B2 phases (Fig. 1b), we can see peak broadening occurs after HPT due to grain refinement and lattice distortion. The Backscattered Electron (BSE) image (Fig. 1c) displays how the B2 phase endured severe strain without a significant reduction in size after HPT[23]. Transmission Electron Microscopy (TEM) investigation with a corresponding Selected Area Electron Diffraction (SAED) pattern of the as-HPT specimen demonstrates the presence of a nanostructured FCC phase (Fig. 1d and Supplementary Fig. 1) as well as relatively fine B2 phase. Energy-Dispersive Spectroscopy (EDS) analysis reveals that the B2 phase is rich in Al and Ni. During the HPT process, the soft FCC grains undergo severe strain, which leads to significant grain refinement, and at the same time, the hard B2 phase flows alongside the soft FCC phase and retains its size[23]. This leads to a unique dual microstructure of nano-sized FCC matrix with a B2 phase in the range of a few hundred nanometers to micrometers.

**Superplastic behavior**. Figure 2a displays the representative engineering stress-strain curves under different testing conditions. The specimens fractured during tensile tests are shown in Fig. 2b and Supplementary Fig. 2. Initially, elongation to failure lengths enhanced proportionally with increasing temperature with the elongation at 873 K being just 430%. Then, superplastic elongation suddenly soared up to 1260% at 973 K. The maximum elongation reached during our tests was 2000% at a temperature of 1073 K under a strain rate of $5 \times 10^{-2}$ s$^{-1}$. Tensile deformed specimens at temperatures of 973 and 1073 K proceeded uniformly without any necking seen at any of the tested strain rates (Fig. 2b and Supplementary Fig. 2). The strain-rate sensitivity ($m$) is a critical parameter used to characterize the ability of metals to resist necking during the deformation. The strain-rate sensitivity can be defined by $m = \partial \ln \sigma / \partial \ln \dot{\varepsilon}$, where $\sigma$ and $\dot{\varepsilon}$ denote the flow stress and plastic strain-rate, respectively. This is shown in Fig. 2c, where the squared R indicates the coefficient of determination. The estimated $m$ value increases with increasing temperature and is very close to the ideal value of 0.5 required for superplasticity at 1073 K[1]. Higher elongations at 973 K and 1073 K, and relative lower elongation at 673 K could be attributed to the estimated $m$ values at these temperatures because high $m$ can decelerate the necking and assist homogeneous deformation[25-27]. The high value of $m$ also implies that the dominant mechanism of superplastic deformation is associated with GBS[1,28,29]. Besides, HPT processing can produce nanoscale grains with a high fraction of high-angle boundaries, thus enabling GBS to occur easily[30-32]. The presence of quasi-stable flow is numerically verified by this $m$ value above and beyond the visual proof of the homogenous elongated specimen's gauge length.

To the best of our knowledge, the elongation to failure result of 2000% achieved in this work is the largest elongation ever reported for HEA and also represents a new record for elongation among the numerous HSRS investigations on conventional alloys[10,33]. Strikingly, the elongation to failure results achieved in this study using a high-strength material is even considered large when compared to conventional superplastic alloys that have low strength at room temperature[24,34-36]. The 2000% elongation in this study far exceeds the 1240% figure reported for wrought AlCoCrCuMnNi HEA[10]. Furthermore, the maximum elongation in this study (2000%) was achieved at a lower temperature (1073 K) than for the AlCoCrCuMnNi HEA (1273 K) and with a far faster

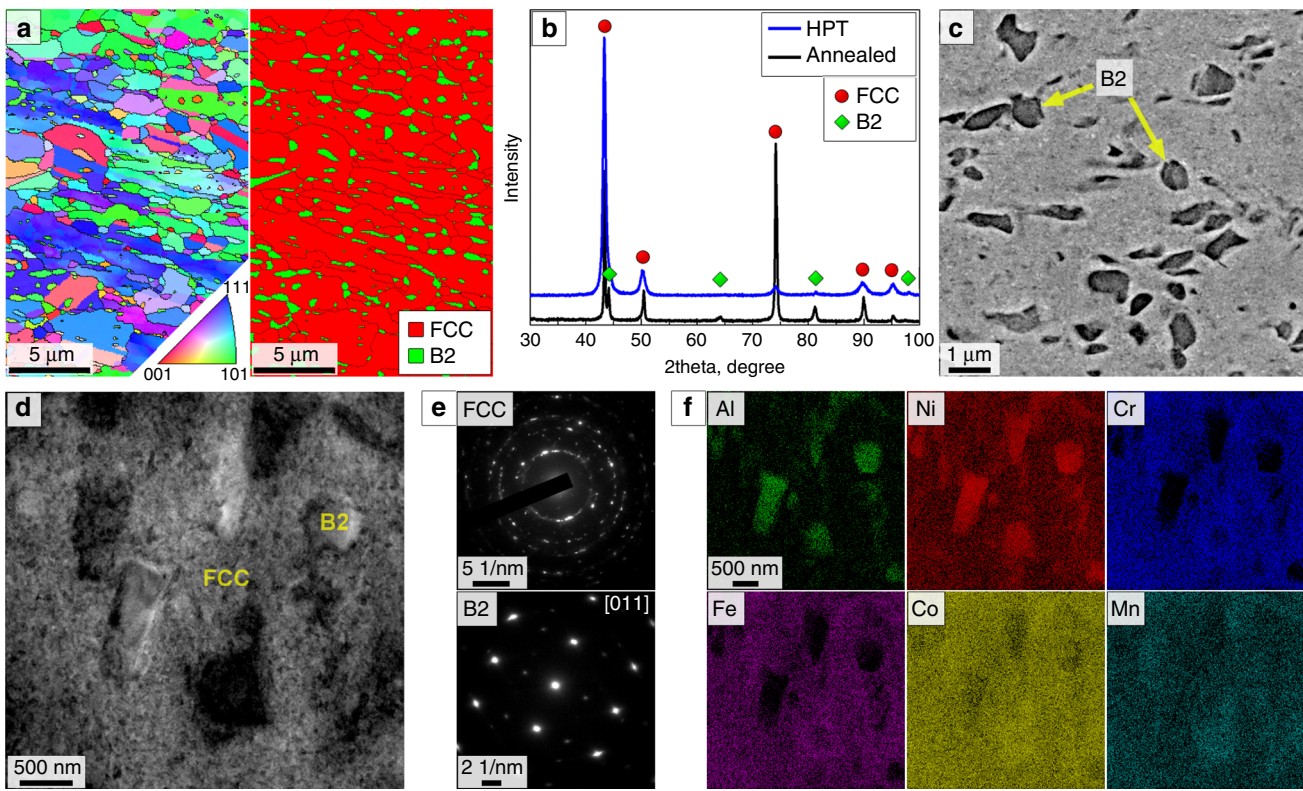

**Fig. 1 Initial microstructure before superplasticity tests. a** Electron Backscatter Diffraction (EBSD)—Inverse Pole Figure (IPF) map and corresponding phase map of annealed specimen, **b** X-Ray Diffraction (XRD) patterns of the annealed and High-Pressure Torsion (HPT) processed specimen, **c** Backscattered Electron (BSE) image of the as-HPT specimen, **d** Transmission Electron Microscopy (TEM) image of the as-HPT specimen, **e** Selected Area Electron Diffraction (SAED) patterns of Face-Centered Cubic (FCC) and ordered Body-Centered Cubic (B2) phases, **f** Transmission Electron Microscopy (TEM)—Energy-Dispersive X-ray Spectroscopy (EDS) maps of the as-HPT specimen showing chemical distribution of the constituent elements.

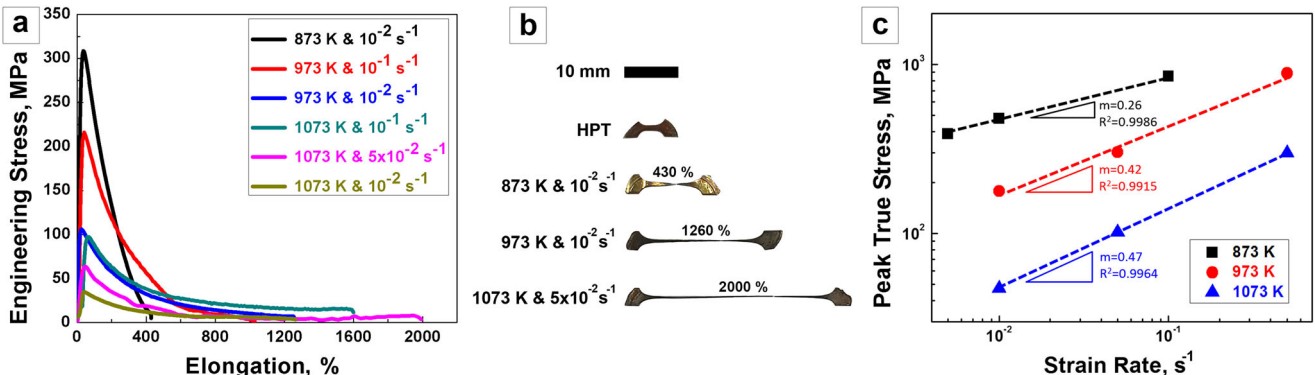

**Fig. 2 Superplastic behavior of Al$_9$(CoCrFeMnNi)$_{91}$ high-entropy alloy (HEA). a** Engineering stress-strain curves of tensile tests at 873, 973, and 1073 K paired with various strain rates from $10^{-1}$ to $10^{-2}$ s$^{-1}$, **b** Images of representative fractured specimens with their testing parameters, **c** Estimated strain-rate sensitivity ($m$) from true stress against strain rate in double logarithmic scale.

strain rate of $5 \times 10^{-2}$ s$^{-1}$ compared to $10^{-2}$ s$^{-1}$ for AlCoCr-CuMnNi. The liquidus temperature of AlCoCrCuMnNi, calculated from CALPHAD, is 1501 K, and this is very similar to 1567 K of the present HEA (Supplementary Fig. 3). The difference in superplastic elongations at 1073–1273 K is notable for these HEAs during thermo-plastic processing[10,37]. From the domain of super-plasticity research that utilizes HPT processing, the present HEA still has the best elongation result compared to the highest

previously reported elongation of 1800% for illustrious Zn-22% Al alloy[38]. It is worth mentioning that the high superplasticity seen in the present work was recorded for a tensile sample with a small cross-sectional area. It has been established in superplasticity literature on severely plastic deformed samples that small cross-sectional area lead to relatively low elongation[33]. For instance, relatively higher elongation has been observed in Zn-22% Al alloy processed by Equal-Channel Angular Pressing (ECAP) compared

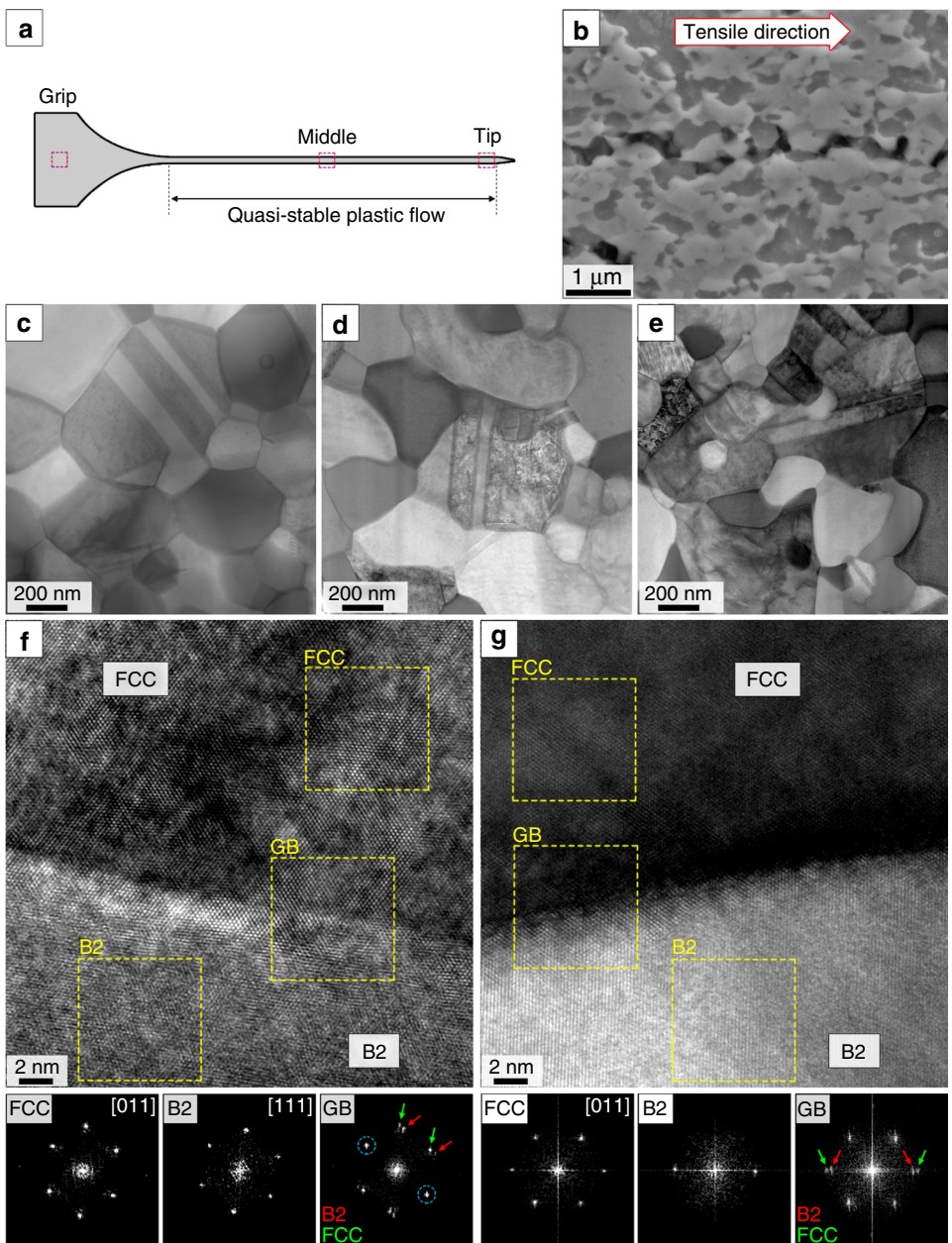

**Fig. 3 Microstructural analysis of superplastic tensile specimen tested at 1073 K and 5 × 10⁻² s⁻¹. a** Schematic illustration of superplasticity specimen with denoted positions for analysis, **b** BSE images of cavity strings from tip of the specimen, **c–e** Scanning Transmission Electron Microscopy (STEM) images from grip, middle and tip, (**f, g**) High Resolution (HR)- TEM images from grip and tip with corresponding Fast Fourier Transform (FFT)- Selected Area Electron Diffraction (SAED) patterns. "GB" in **f**, **g** stands for Grain Boundary.

to HPT-processed samples, this lower elongation in the HPT-processed sample is attributed to its small cross-sectional area compared to the ECAP sample[38].

**Superplasticity mechanism.** Microstructural investigations of the superplastic tensile sample at 2000% elongation from various positions along the deformed specimen are shown in Fig. 3. We can see the cavity strings are in alignment with the applied tension direction (Fig. 3b). Strikingly, the Scanning Transmission Electron Microscopy (STEM) images show that there is no considerable grain size difference between the deformed (gauge) and nondeformed (grip) parts (Fig. 3c–e), this indicates an almost total absence of dynamic grain growth in the deformed region[39]. The TEM-EDS analysis of the specimens tested at 1073 K shows

the formation of sigma phase (rich in Cr) during the superplasticity tests in addition to the presence of FCC and B2 phases (Supplementary Fig. 4). The high-strain-rate and presence of B2 and sigma phases limit grain growth during superplastic deformation, leading to a quasi-theoretical $m$ value. Also, the grain morphology changes from an equiaxed shape at the grip to semi-equiaxed morphology at the tip. High Resolution (HR)- TEM images from the interface of FCC and B2 phases at the grip are shown by Fig. 3f and at the tip are shown by Fig. 3g. Corresponding Fast Fourier Transform (FFT)- generated SAED patterns expose the presence of a Kurdjumov–Sachs (K–S) Orientation Relationship (OR; depicted by the blue circle) at the grip. However, there is no specific OR between FCC and B2 phases at the tip.

Maintaining UFG and near-equiaxed morphology during the superplasticity test confirms the occurrence of GBS[1]. Elimination of OR between FCC and B2 points toward the occurrence of GBS that accommodates grain rotation during the superplasticity test[40]. In addition, the generation of the high dislocation density seen mainly in FCC and B2 grains (Supplementary Fig. 5) along with the evolution of the semi-equiaxed morphology at the tip suggests the role of intragranular dislocations for the accommodation of plastic deformation during the superplasticity test[9,41]. However, the formation of intragranular dislocation in the sigma phase was not observed. The HR-TEM images taken from the interface show a high dislocation density around the interphase boundaries in the tip (Supplementary Figs. 6–8). Looking at the above evidence, we can say the different phases in the present multi-phase HEA demonstrate quite a heterogeneous mechanical response at the scale of individual grains during the superplasticity test. It can be concluded that the sigma phase may play the role of a hard domain in the present HEA and Geometrically Necessary Dislocations (GNDs) can be generated in the softer domains to maintain interphase boundary compatibility[42]. Accordingly, the concurrent process of GBS and GBS being accommodated by intragranular dislocation and the generation of GNDs come together to contribute to the record elongation by postponing the initiation of cracks at interphase boundaries. A detailed survey on the superplasticity of HEAs shows that the accommodation of GBS by dislocation activity is rather limited[43].

**Cavity interlinkage.** Cavity damage due to GBS can be restricted by the rare interlinkage of cavity strings under plasticity-controlled cavity growth[21,44]. The width of cavity strings is on the order of grain size and originates at the interphase of FCC and B2 phases[9] (Supplementary Fig. 9). Cavities tend to grow along FCC grains parallel to the tensile direction and get blocked by B2 and sigma grains[4,45]. The ease of cavity interlinkage is manipulated by the interleaved microstructure of the present HEA, and this has the crucial impact of making it easy to retain quasi-stable plastic flow[46]. In HSRS, partial melting is suggested as one of the accommodation mechanisms to assist GBS due to the formation of a liquid-phase film at the boundaries[47,48]. However, TEM-EDS line-scanning at different interphase boundaries at the tip revealed a sharp change in the chemical compositions without any evidence of partial melting (i.e., segregation of an element or formation of an intermediate phase) at the boundaries in the present HEA (Supplementary Fig. 10).

In summary, this work shows that it is possible to achieve ultra-high HSRS in HEA through microstructural engineering. The formation of dual-phase microstructure during HPT and the subsequent formation of a sigma phase led to a multi-phase UFG microstructure during deformation. This UFG microstructure improves GBS efficiency leading to steady-state plastic flow and an estimated $m$ value close to the ideal value. In addition, the interlinkage of cavities and the dislocation motion in FCC and B2 phases reduce stress concentrations and play a vital role in accommodating GBS. Consequently, the present multi-phase microstructure is able to achieve an ultra-high HSRS with a the record of 2000% elongation that is significantly greater than anything seen in previous investigations from the severe plastic deformation society and HEA community. In addition, the unique properties of HEAs together with this ultra-high HSRS make HEAs even more attractive for aerospace and automobile industries. We propose that the practical improvement of superplasticity in HEA materials can be obtained by taking advantage of a multi-phase UFG structure.

## Methods

**Sample preparation and superplasticity tensile tests.** The $Al_9(CoCrFeMnNi)_{91}$ (at%) HEA was produced from elements with a purity of 99.99% by vacuum induction melting. The as-cast ingot was homogenized at 1473 K for 12 h, followed by water quenching. The ingot was subjected to cold-rolling to reduce its thickness from 7 to 1.5 mm, and this corresponds to a 78.6% reduction. The cold-rolled ingot was annealed at 1273 K for 15 min in argon gas, followed by water quenching. Discs with a diameter of 10 mm were machined from the annealed plate through electric discharge machining. The 10 mm discs were processed by five revolutions of HPT at room temperature under an applied pressure of 6 GPa with an anvil rotation rate of 1 revolution per minute.

Differential Scanning Calorimetry (DSC) measurements, using the SETARAM LABSYS evo DTA/DSC instrument, were conducted on a piece of the HPT-processed sample (~100 mg). The HPT sample was loaded into a ceramic pan and then heated from room temperature to melting temperature with a heating rate of 10 K/min in a nitrogen gas atmosphere. The flow rate of the $N_2$ gas was controlled to remain at 50 mL/min. The absolute melting temperature from DSC was employed to determine the testing temperatures used in the superplasticity experiments.

Flat dog-bone-shaped specimens were machined symmetrically from the as-HPT disks with the gauge's position set at 2.5 mm away from the center. The gauge dimensions are 1.5 mm in length, 1.0 mm in width, and 0.7 mm in thickness. The specimens were firmly assembled to a pair of nonslippery clamps at the grip parts by nuts, then subjected to tension tests using an Instron 8862 instrument. The specimens were well aligned to be parallel with the pulling shaft. The specimens were heated from room temperature to the target superplastic testing temperatures of 873, 973, and 1073 K by halogen furnace (DF-60HG, DaeHeung Co., Korea) in the air atmosphere. After the stabilization of temperature with a dwell time of 7 min, the specimens were preloaded until 5 N to remove the thermal expansion effect of the system. Then, a traction load was applied until the fracture. The failure specimens were promptly immersed in water to preserve their microstructure. Three tests were carried out in each condition to ensure the reproducibility of results. The elongation was measured using the recorded displacement from the Instron data with a step size of 1 s on each different strain rate. In addition, the obtained elongation values were confirmed by evaluating the superplastic fractured samples using an optical microscope.

**Microstructural characterization.** The Transmission Electron Microscopy (TEM) specimens of the as-HPT and grip of the superplastic specimens were prepared by mechanical grinding to a thickness of ~60 μm, and they were then thinned using electro-polishing with a solution of 10% $HClO_4$ and 90% $CH_3COOH$ at an applied voltage of 25 V at room temperature. The TEM foils from the middle and tip of superplastic specimens were carried out using the Focused Ion Beam (FIB) lift-out procedure. The TEM analysis was undertaken by the JEM-2100F (JEOL, Japan) instrument.

The surface of specimens was polished to a mirror-like finish using diamond paste and colloidal silica suspension for the following analysis.

Phase identification was determined by X-Ray Diffraction (XRD) using the Rigaku D/MAX-2500 XRD instrument employing Cu Kα radiation with a wavelength of 0.154 nm. The XRD pattern was acquired in a 2θ range of 30–100° at a scan rate of 1° per min and a scan-interval of 0.02°.

The microstructure of the annealed specimen was observed using Electron Backscatter Diffraction (EBSD), performed by a Field Emission-Scanning Electron Microscope (FE-SEM; Quanta 3D, FEI Company, USA). The obtained data were analyzed using Orientation Imaging Microscopy (OIM) analysis software (TSL OIM analysis 7).

## Data availability

The data that support the findings of this study are available from the corresponding author upon reasonable request.

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

## Acknowledgements
This investigation was supported by the Creative Materials Discovery Program through the National Research Foundation of Korea (NRF) granted by the Ministry of Science and ICT (2016M3D1A1023384). P.S. was funded by the Korea Research Fellowship program through the National Research Foundation of Korea (NRF) sponsored by the Ministry of Science and ICT (2017H1D3A1A01013666). We have gratefully received the advisory from Dr. Jongun Moon from POSTECH and Dr. Hamed Shahmir from the University of Sheffield, England.

## Author contributions
N.T-C.N. and P.A-R. contributed equally to this work. N.T-C.N. generated the idea of superplastic alloy component and testing conditions, performed all experiments, processed data, and wrote the manuscript. P.A-R. conducted the sample preparation for superplastic tests (casting, rolling, HPT), performed the microstructural analysis using TEM and EBSD, and contributed to the writing and discussion part. S.P. supported the BSE and FIB implementation, revised the text, and contributed to the discussion. A.Z. contributed to the discussion part and accomplished the TEM works. C.S.L. provided the technical instruction for entire superplastic tensile tests. H.S.K. organized the work, supervised the research progress, and finalized the manuscript.

## Competing interests
The authors declare no competing interests.
