## [Peer Review File · Nature Communications]

Reviewers' Comments:

Reviewer #1:

Remarks to the Author:

The authors presented remarkable results regarding exceptional superplastic behavior of the so-called high-entropy alloy. Proposed results were obtained due to nanostructuring of the Al₉(CoCrFeMnNi)₉₁ alloy by means of high-pressure torsion and subsequent studies of mechanical properties (tensile tests) at high temperatures (up to 1073 K). Investigation of high-entropy alloys is one of the most topical themes during last decade in materials science and might be especially interesting for potential Reader of Nature Communications. The authors found an attractive application of high-pressure torsion technique for the noted alloy composition and fulfilled comprehensive measurements of mechanical characteristics and TEM observations including high resolution. Accuracy in the comparative analysis of microstructural data seems to be a good opportunity to publish this manuscript.

At the same time, there are some minor points that could be improved to make content more clear:

1. According to existing data, effect of Al addition was already studied in the case of compositions, which are close to Cantor type of high-entropy alloy (CoCrFeMnNi). Therefore, reasoning of the choice of the alloy should be actual for this study.
2. Flat dog-bone shaped specimens cut from the HPT disks usually possess relatively small gauge dimensions (1.5 mm in length, 1.0 mm in width and 0.7 mm in thickness here) in comparison with standard samples for tensile tests. That is why, the methodical part concerning the accuracy of the elongation control should be added.
3. Page 6, rows 113-115. The sentence regarding "crystal lattice distortions produced by HPT process have been completely reconstructed at 973 and 1073 K" seems to be confusing because the recovery and grain growth were started at much lower temperatures.

Reviewer #2:

Remarks to the Author:

For their work, the authors investigated the large plastic deformation (deemed superplasticity) exhibited by a Al₉(CoCrFeMnNi)₉₁ (at%) high entropy alloy (HEA) using High-Pressure Torsion (HPT). This work appears well put together and with a few minor corrections it should be acceptable for publication.

1. This alloy contains Al and consists of a face centered cubic and Al-Ni rich B2 phase in the as-cast condition. Please give a more detailed discussion of their significance in HEAs. Below are a few references to help with this process:

S. Chen, X. Xie, W. Li, R. Feng, B. Chen, J. Qiao, Y. Ren, Y. Zhang, K.A. Dahmen, P.K. Liaw, Temperature effects on the serrated behavior of an Al_{0.5}CoCrCuFeNi high-entropy alloy, Mater. Chem. Phys. 210 (2018) 20-28.

Y. Zhang, T.T. Zuo, Z. Tang, M.C. Gao, K.A. Dahmen, P.K. Liaw, Z.P. Lua, Microstructures and properties of high-entropy alloys, Prog. Mater. Sci. 61 (2014) 1-93.

2. Please provide a reference for the following statement:

"Cavities tend to grow along FCC grains and get blocked by B2 and sigma grains."

3. Were the torsion tests performed in air?

April 30, 2020

Response Letter

Ref. No.: NCOMMS-20-06973-T

Title: Exceptional high-strain rate superplasticity in high-entropy alloy

Dear Reviewer,

Thank you for your constructive review and comments for our manuscript. We realized that our previous manuscript has some mistakes. We have addressed this problem in the revised manuscript.

In the revised manuscript, the revised parts were highlighted in yellow, and some parts revised additionally were highlighted in green.

We hope that our answers are satisfactory and clear.

Best regards,

All authors

Response to reviewer's comments

Reviewer #1:

The authors presented remarkable results regarding exceptional superplastic behavior of the so-called high-entropy alloy. Proposed results were obtained due to nanostructuring of the Al₉(CoCrFeMnNi)₉₁ alloy by means of high-pressure torsion and subsequent studies of mechanical properties (tensile tests) at high temperatures (up to 1073 K). Investigation of high-entropy alloys is one of the most topical themes during last decade in materials science and might be especially interesting for potential Reader of Nature Communications. The authors found an attractive application of high-pressure torsion technique for the noted alloy composition and fulfilled comprehensive measurements of mechanical characteristics and TEM observations including high resolution. Accuracy in the comparative analysis of microstructural data seems to be a good opportunity to publish this manuscript.

At the same time, there are some minor points that could be improved to make content more clear:

Comment #1:

According to existing data, effect of Al addition was already studied in the case of compositions, which are close to Cantor type of high-entropy alloy (CoCrFeMnNi). Therefore, reasoning of the choice of the alloy should be actual for this study.

Response #1:

We would like to thank the reviewer for the valuable suggestion. This comment was addressed as follows:

“The representative CoCrFeMnNi HEA with a single Face-Centered Cubic (FCC) structure exhibited a reasonable superplastic elongation of 570% at elevated temperature¹⁵. There is a consensus that a multi-phase structure potentially shows higher elongation as compared with a single-phase structure due to grain growth suppression during the superplasticity test^{1,7}. In most of the HEAs, the addition of Al has been demonstrated to form an ordered Body-Centered Cubic (B2) phase and it is shown to enhance the room temperature mechanical properties¹⁶⁻¹⁸. It has been reported that the B2 phase in Al_x(CoCrFeMnNi)_{100-x} forms when the Al content is above 8 at%, and a drastic decrease in ductility is observed when the Al content is more than 11 at%¹⁹. Consequently, in the present study, we chose Al₉CoCrFeMnNi HEA based on the abovementioned logical reasoning and engineered a unique nanostructured dual-phase microstructure through thermo-mechanical treatment followed by HPT processing.”

Comment #2:

Flat dog-bone shaped specimens cut from the HPT disks usually possess relatively small gauge dimensions (1.5 mm in length, 1.0 mm in width and 0.7 mm in thickness here) in comparison with standard samples for tensile tests. That is why, the methodical part concerning the accuracy of the elongation control should be added.

Response #2:

We would like to appreciate this valuable comment from the reviewer. We have modified the “Methods” section as follows:

“Flat dog-bone shaped specimens were machined symmetrically from the as-HPT disks with the gauge’s position set at 2.5 mm away from the center. The gauge dimensions are 1.5 mm in length, 1.0 mm in width, and 0.7 mm in thickness. The specimens were firmly assembled to a pair of non-slippery clamps at the grip parts by nuts, then subjected to tension tests using an Instron 8862 instrument. The specimens were well aligned to be parallel with the pulling shaft. The specimens were heated from room temperature to the target superplastic testing temperatures of 873, 973, and 1073 K by halogen furnace (DF-60HG, DaeHeung Co., Korea) in the air atmosphere. After the stabilization of temperature with a dwell time of 7 minutes, the specimens were preloaded until 5 N to remove the thermal expansion effect of the system. Then, a traction load was applied until the fracture happened. The failure specimens were promptly immersed in water to preserve their microstructure. Three tests were carried out in each condition to ensure the reproducibility of results. The elongation was measured using the recorded displacement from the Instron data with a step size of 1 second on each different strain rate. In addition, the obtained elongation values were confirmed by evaluating the superplastic fractured samples using an optical microscope.”

Comment #3:

Page 6, rows 113-115. The sentence regarded “crystal lattice distortions produced by HPT process have been completely reconstructed at 973 and 1073 K” seems to be confusing because the recovery and grain growth were started at much lower temperatures.

Response #3:

We would like to thank the reviewer for the careful review. We totally agree with the reviewer that that statement was confusing. We modified the manuscript as follows:

“Higher elongations at 973 K and 1073 K, and relative lower elongation at 673 K could be attributed to the estimated m values at these temperatures because high m can decelerate the necking and assist homogeneous deformation²²⁻²⁴. The high value of m also implies that the dominant mechanism of superplastic deformation is associated with GBS^{1,25,26}. Besides, HPT

processing can produce nano-scale grains with a high fraction of high-angle boundaries, thus enabling GBS to occur easily²⁷⁻²⁹ .”

Reviewer #2:

For their work, the authors investigated the large plastic deformation (deemed superplasticity) exhibited by a Al₉(CoCrFeMnNi)₉₁ (at%) high entropy alloy (HEA) using High-Pressure Torsion (HPT). This work appears well put together and with a few minor corrections it should be acceptable for publication.

Comment #1:

This alloy contains Al and consists of a face centered cubic and Al-Ni rich B2 phase in the as-cast condition. Please give a more detailed discussion of their significance in HEAs. Below are a few references to help with this process:

S. Chen, X. Xie, W. Li, R. Feng, B. Chen, J. Qiao, Y. Ren, Y. Zhang, K.A. Dahmen, P.K. Liaw, Temperature effects on the serrated behavior of an Al_{0.5}CoCrCuFeNi high-entropy alloy, *Mater. Chem. Phys.* 210 (2018) 20-28.

Y. Zhang, T.T. Zuo, Z. Tang, M.C. Gao, K.A. Dahmen, P.K. Liaw, Z.P. Lua, Microstructures and properties of high-entropy alloys, *Prog. Mater. Sci.* 61 (2014) 1-93.

Response #1:

We thank the reviewer for the valuable suggestion to discuss and cite the important articles. As suggested by the reviewer, we have modified the manuscript as follows:

“The representative CoCrFeMnNi HEA with a single Face-Centered Cubic (FCC) structure exhibited a reasonable superplastic elongation of 570% at elevated temperature¹⁵. There is a consensus that a multi-phase structure potentially shows higher elongation as compared with a single-phase structure due to grain growth suppression during the superplasticity test^{1,7}. In most of the HEAs, the addition of Al has been demonstrated to form an ordered Body-Centered Cubic (B2) phase and it is shown to enhance the room temperature mechanical properties¹⁶⁻¹⁸. It has been reported that the B2 phase in Al_x(CoCrFeMnNi)_{100-x} forms when the Al content is above 8 at%, and a drastic decrease in ductility is observed when the Al content is more than 11 at%¹⁹. Consequently, in the present study, we chose Al₉CoCrFeMnNi HEA based on the abovementioned logical reasoning and engineered a unique nanostructured dual-phase microstructure through thermo-mechanical treatment followed by HPT processing.”

And the suggested informative references are added to the manuscript:

“[13] Zhang, Y. et al. Microstructures and properties of high-entropy alloys. *Prog. Mater. Sci.* 61, 1-93 (2014).”

“[17] Chen, S. et al. Temperature effects on the serrated behavior of an Al_{0.5}CoCrCuFeNi high-entropy alloy. *Mater. Chem. Phys.* 210, 20-28 (2018).”

Comment #2:

Please provide a reference for the following statement:

"Cavities tend to grow along FCC grains and get blocked by B2 and sigma grains."

Response #2:

We thank the reviewer for the careful review, and we have modified the manuscript and added the reference as follows:

“Cavities tend to grow along FCC grains parallel to the tensile direction and get blocked by B2 and sigma grains^{4,43}.”

“[4] Park, K.T., Hwang, D.Y., Lee, Y.K., Kim, Y.D. & Shin, D.H. High strain rate superplasticity of submicrometer grained 5083 Al alloy containing scandium fabricated by severe plastic deformation. *Mater. Sci. Eng. A* 341, 273-281 (2003).”

“[43] Leng, Z. et al. Superplastic behavior of extruded Mg–9RY–4Zn alloy containing long period stacking ordered phase. *Mater. Sci. Eng. A* 576, 202-206 (2013).”

Comment #3:

Were the torsion tests performed in air?

Response #3:

We thank the reviewer for raising this concern regarding the testing atmosphere.

The high-pressure torsion process was conducted in the air atmosphere at room temperature.

Also, the superplastic tensile tests were performed in the air atmosphere with the heating generated from the halogen furnace. The Al₉(CoCrFeMnNi)₉₁ HEA presented a reasonable oxidation resistance; therefore, no special gas protection solution was employed during the superplasticity tests. The manuscript is modified as follows:

“The specimens were heated from room temperature to the target superplastic testing temperatures of 873, 973, and 1073 K by halogen furnace (DF-60HG, DaeHeung Co., Korea) in the air atmosphere.”